# THE ROLE OF PRETRAINED REPRESENTATIONS FOR THE OOD GENERALIZATION OF RL AGENTS

**Frederik Träuble,**[*,1] **Andrea Dittadi,**[*,1,2] **Manuel Wüthrich,**[1] **Felix Widmaier,**[1] **Peter Gehler,**[3]
**Ole Winther,**[2] **Francesco Locatello,**[3] **Olivier Bachem,**[4] **Bernhard Schölkopf,**[1] **Stefan Bauer**[1,5]

[1]Max Planck Institute for Intelligent Systems, Tübingen, Germany,
[2]Technical University of Denmark, [3]Amazon Lablets, [4]Google Brain, [5]KTH Stockholm

## ABSTRACT

Building sample-efficient agents that generalize out-of-distribution (OOD) in real-world settings remains a fundamental unsolved problem on the path towards achieving higher-level cognition. One particularly promising approach is to begin with low-dimensional, pretrained representations of our world, which should facilitate efficient downstream learning and generalization. By training 240 representations and over 10,000 reinforcement learning (RL) policies on a simulated robotic setup, we evaluate to what extent different properties of pretrained VAE-based representations affect the OOD generalization of downstream agents. We observe that many agents are surprisingly robust to realistic distribution shifts, including the challenging sim-to-real case. In addition, we find that the generalization performance of a simple downstream proxy task reliably predicts the generalization performance of our RL agents under a wide range of OOD settings. Such proxy tasks can thus be used to select pretrained representations that will lead to agents that generalize.

## 1 INTRODUCTION

Robust out-of-distribution (OOD) generalization is one of the key open challenges in machine learning. This is particularly relevant for the deployment of ML models to the real world, where we need systems that generalize beyond the i.i.d. (independent and identically distributed) data setting (Schölkopf et al., 2021; Djolonga et al., 2020; Koh et al., 2021; Barbu et al., 2019; Azulay & Weiss, 2019; Roy et al., 2018; Gulrajani & Lopez-Paz, 2020; Hendrycks & Dietterich, 2019; Michaelis et al., 2019; Funk et al., 2021). One instance of such models are agents that learn by interacting with a training environment and we would like them to generalize to other environments with different statistics (Zhang et al., 2018; Pfister et al., 2019; Cobbe et al., 2019; Ahmed et al., 2021; Ke et al., 2021). Consider the example of a robot with the task of moving a cube to a target position: Such an agent can easily fail as soon as some aspects of the environment differ from the training setup, e.g. the shape, color, and other object properties, or when transferring from simulation to real world.

Humans do not suffer from these pitfalls when transferring learned skills beyond a narrow training domain, presumably because they represent visual sensory data in a concise and useful manner (Marr, 1982; Gordon & Irwin, 1996; Lake et al., 2017; Anand et al., 2019; Spelke, 1990). Therefore, a particularly promising path is to base predictions and decisions on similar low-dimensional representations of our world (Bengio et al., 2013; Kaiser et al., 2019; Finn et al., 2016; Barreto et al., 2017; Dittadi et al., 2021a; Stooke et al., 2021; Vinyals et al., 2019). The learned representation should facilitate efficient downstream learning (Eslami et al., 2018; Anand et al., 2019; Stooke et al., 2021; Van Steenkiste et al., 2019) and exhibit better generalization (Zhang et al., 2020; Srinivas et al., 2020). Learning such a representation from scratch for every downstream task and every new variation would be inefficient. If we learned to juggle three balls, we should be able to generalize to oranges or apples without learning again from scratch. We could even do it with cherimoyas, a fruit that we might have never seen before. We can effectively reuse our generic representation of the world.

---

[*]Equal contribution. The order was chosen at random and can be swapped. Correspondence to: frederik.traeuble@tuebingen.mpg.de and adit@dtu.dk.

We thus consider deep learning agents trained from pretrained representations and ask the following questions: To what extent do they generalize under distribution shifts similar to those mentioned above? Do they generalize in different ways or to different degrees depending on the type of distribution shift, including sim-to-real? Can we predict the OOD generalization of downstream agents from properties of the pretrained representations?

To answer the questions above, we need our experimental setting to be realistic, diverse, and challenging, but also controlled enough for the conclusions to be sound. We therefore base our study on the robot platform introduced by Wüthrich et al. (2020). The scene comprises a robot finger with three joints that can be controlled to manipulate a cube in a bowl-shaped stage. Dittadi et al. (2021c) conveniently introduced a dataset of simulated and real-world images of this setup with ground-truth labels, which can be used to pretrain and evaluate representations. To train downstream agents, we adapted the simulated reinforcement learning benchmark CausalWorld from Ahmed et al. (2021) that was developed for this platform. Building upon these works, we design our experimental study as follows (see Fig. 1): First, we pretrain representations from static simulated images of the setup and evaluate a collection of representation metrics. Following prior work (Watter et al., 2015; Van Hoof et al., 2016; Ghadirzadeh et al., 2017; Nair et al., 2018; Ha & Schmidhuber, 2018; Eslami et al., 2018), we focus on autoencoder-based representations. Then, we train downstream agents from this fixed representation on a set of environments. Finally, we investigate the zero-shot generalization of these agents to new environments that are out of the training distribution, including the real robot.

The goal of this work is to provide the first systematic and extensive account of the OOD generalization of downstream RL agents in a robotic setup, and how this is affected by characteristics of the upstream pretrained representations. We summarize our contributions as follows:

- We train 240 representations and 11,520 downstream policies,[1] and systematically investigate their performance under a diverse range of distribution shifts.[2]
- We extensively analyze the relationship between the generalization of our RL agents and a substantial set of representation metrics.
- Notably, we find that a specific representation metric that measures the generalization of a simple downstream proxy task reliably predicts the generalization of downstream RL agents under the broad spectrum of OOD settings considered here. This metric can thus be used to select pretrained representations that will lead to more robust downstream policies.
- In the most challenging of our OOD scenarios, we deploy a subset of the trained policies to the corresponding real-world robotic platform, and observe surprising zero-shot sim-to-real generalization without any fine-tuning or domain randomization.

## 2 BACKGROUND

In this section, we provide relevant background on the methods for representation learning and reinforcement learning, and on the robotic setup to evaluate out-of-distribution generalization.

**Variational autoencoders.** VAEs (Kingma & Welling, 2014; Rezende et al., 2014) are a framework for optimizing a latent variable model $p_\theta(\mathbf{x}) = \int_{\mathbf{z}} p_\theta(\mathbf{x} \mid \mathbf{z}) p(\mathbf{z}) d\mathbf{z}$ with parameters $\theta$, typically with a fixed prior $p(\mathbf{z}) = \mathcal{N}(\mathbf{z}; \mathbf{0}, \mathbf{I})$, using amortized stochastic variational inference. A variational distribution $q_\phi(\mathbf{z} \mid \mathbf{x})$ with parameters $\phi$ approximates the intractable posterior $p_\theta(\mathbf{z} \mid \mathbf{x})$. The approximate posterior and generative model, typically called encoder and decoder and parameterized by neural networks, are jointly optimized by maximizing a lower bound to the log likelihood (the ELBO):

$$\log p_\theta(\mathbf{x}) \geq \mathbb{E}_{q_\phi(\mathbf{z} \mid \mathbf{x})} \left[ \log p_\theta(\mathbf{x} \mid \mathbf{z}) \right] - D_{\mathrm{KL}} \left( q_\phi(\mathbf{z} \mid \mathbf{x}) \| p(\mathbf{z}) \right) = \mathcal{L}_{\theta, \phi}^{ELBO}(\mathbf{x}) \,. \tag{1}$$

In $\beta$-VAEs, the KL term is modulated by a factor $\beta$ to enforce a more structured latent space (Higgins et al., 2017a; Burgess et al., 2018). While VAEs are typically trained without supervision, we also employ a form of weak supervision (Locatello et al., 2020) that encourages disentanglement.

**Reinforcement learning.** A Reinforcement Learning (RL) problem is typically modeled as a Partially Observable Markov Decision Process (POMDP) defined as a tuple $(S, A, T, R, \Omega, O, \gamma, \rho_0, H)$

---

[1] Training the representations required approximately 0.62 GPU years on NVIDIA Tesla V100. Training and evaluating the downstream policies required about 86.8 CPU years on Intel Platinum 8175M.

[2] Additional results and videos are provided at `https://sites.google.com/view/ood-rl`.

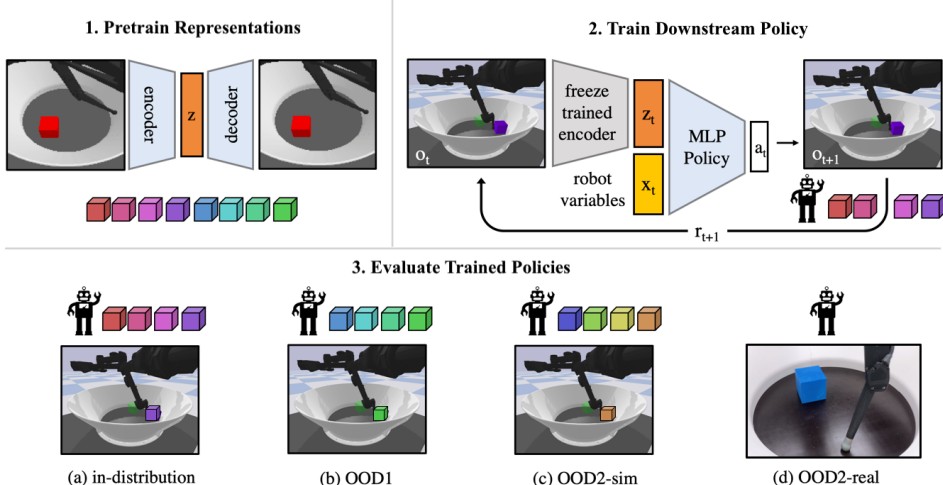

Figure 1: **Overview of our experimental setup for investigating out-of-distribution generalization in down-stream tasks.** (1) We train 240 $\beta$-VAEs on the robotic dataset from Dittadi et al. (2021c). (2) We then train downstream policies to solve *object reaching* or *pushing*, using multiple random RL seeds per VAE. The input to a policy consists of the output of a pretrained encoder and additional task-related observable variables. Crucially, the policy is only trained on a subset of the cube colors from the pretraining dataset. (3) Finally, we evaluate these policies on their respective tasks in four different scenarios: (a) in-distribution, i.e. with cube colors used in policy training; (b) OOD1, i.e. with cube colours previously seen by the encoder but OOD for the policy; (c) OOD2-sim, having cube colours also OOD to the encoder; (d) sim-to-real zero-shot on the real-world setup.

with states $s \in S$, actions $a \in A$ and observations $o \in \Omega$ determined by the state and action of the environment $O(o|s, a)$. $T(s_{t+1}|s_t, a_t)$ is the transition probability distribution function, $R(s_t, a_t)$ is the reward function, $\gamma$ is the discount factor, $\rho_0(s)$ is the initial state distribution at the beginning of each episode, and $H$ is the time horizon per episode. The objective in RL is to learn a policy $\pi : S \times A \to [0, 1]$, typically parameterized by a neural network, that maximizes the total discounted expected reward $J(\pi) = \mathbb{E}\left[\sum_{t=0}^{H} \gamma^t R(s_t, a_t)\right]$. There is a broad range of model-free learning algorithms to find $\pi^*$ by policy gradient optimization or by learning value functions while trading off exploration and exploitation (Haarnoja et al., 2018b; Schulman et al., 2017; Sutton et al., 1999; Schulman et al., 2015b;a; Silver et al., 2014; Fujimoto et al., 2018). Here, we optimize the objective above with *Soft Actor Critic* (SAC), an off-policy method that simultaneously maximizes the expected reward and the entropy $H(\pi(\cdot|s_t))$, and is widely used in control tasks due to its sample efficiency (Haarnoja et al., 2018b).

**A robotic setup to evaluate out-of-distribution generalization.** Our study is based on a real robot platform where a robotic finger with three joints manipulates a cube in a bowl-shaped stage (Wüthrich et al., 2020). We pretrain representations on a labeled dataset introduced by Dittadi et al. (2021c) which consists of simulated and real-world images of this setup. This dataset has 7 underlying factors of variation (FoV): angles of the three joints, and position (x and y), orientation, and color of the cube. Some of these factors are correlated (Dittadi et al., 2021c), which may be problematic for representation learners, especially in the context of disentanglement (Träuble et al., 2021; Chen et al., 2021). After training the representations, we train downstream agents and evaluate their generalization on an adapted version of the simulated CausalWorld benchmark (Ahmed et al., 2021) that was developed for the same setup. Finally, we test sim-to-real generalization on the real robot.

Our experimental setup, illustrated in Fig. 1, allows us to systematically investigate a broad range of out-of-distribution scenarios in a controlled way. We pretrain our representations from this simulated dataset that covers 8 distinct cube colors. We then train an agent from this fixed representation on a subset of the cube colors, and evaluate it (1) on the same colors (this is the typical scenario in RL), (2) on the held-out cube colors that are still known to the encoder, or (3) OOD w.r.t. the encoder's training distribution, e.g. on novel colors and shapes or on the real world.

We closely follow the framework for measuring OOD generalization proposed by Dittadi et al. (2021c). In this framework, a representation is initially learned on a training set $\mathcal{D}$, and a simple downstream model is trained on a subset $\mathcal{D}_1 \subset \mathcal{D}$ to predict the ground-truth factors from the learned representation. Generalization is then evaluated by testing the downstream model on a set $\mathcal{D}_2$

that differs distributionally from $\mathcal{D}_1$, e.g. containing images corresponding to held-out values of a chosen factor of variation (FoV). Dittadi et al. (2021c) consider two flavors of OOD generalization depending on the choice of $\mathcal{D}_2$: First, the case when $\mathcal{D}_2 \subset \mathcal{D}$, i.e. the OOD test set is a subset of the dataset for representation learning. This is denoted by **OOD1** and corresponds to the scenario (2) from the previous paragraph. In the other scenario, referred to as **OOD2**, $\mathcal{D}$ and $\mathcal{D}_2$ are disjoint and distributionally different. This even stronger OOD shift corresponds to case (3) above. The generalization score for $\mathcal{D}_2$ is then measured by the (normalized) mean absolute prediction error across all FoVs except for the one that is OOD. Following Dittadi et al. (2021c), we use a simple 2-layer Multi-Layer Perceptron (MLP) for downstream factor prediction, we train one MLP for each FoV, and report the *negative* error. This simple and cheap generalization metric could serve as a convenient proxy for the generalization of more expensive downstream tasks. We refer to these generalization scores as GS-OOD1, GS-OOD2-sim, and GS-OOD2-real depending on the scenario.

The focus of Dittadi et al. (2021c) was to scale VAE-based approaches to more realistic scenarios and study the generalization of these simple downstream tasks, with a particular emphasis on disentanglement. Building upon their contributions, we can leverage the broader potential of this robotic setup with many more OOD2 scenarios to study our research questions: To what extent can agents generalize under distribution shift? Do they generalize in different ways depending on the type of shift (including sim-to-real)? Can we predict the OOD generalization of downstream agents from properties of the pretrained representations such as the GS metrics from Dittadi et al. (2021c)?

## 3 STUDY DESIGN

**Robotic setup.** Our setup is based on TriFinger (Wüthrich et al., 2020) and consists of a robotic finger with three joints that can be controlled to manipulate an object (e.g. a cube) in a bowl-shaped stage. The agent receives a camera observation consistent with the images in Dittadi et al. (2021c) and outputs a three-dimensional action. During training, which always happens in simulation, the agent only observes a cube of four possible colors, randomly sampled at every episode (see Fig. 1, step 2).

**Distribution shifts.** After training, we evaluate these agents in 7 environments: (1) the training environment, which is the typical setting in RL, (2) the OOD1 setting with cube colors that are OOD for the agent but still in-distribution for the encoder, (3) the more challenging OOD2-sim setting where the colors are also OOD for the encoder, (4-6) the OOD2 settings where the object colors are as in the 3 previous settings but the cube is replaced by a sphere (a previously unseen shape), (7) the OOD2-real setting, where we evaluate zero-shot sim-to-real transfer on the real robotic platform.

**Tasks.** We begin our study with the *object reaching* downstream control task, where the agent has to reach an object placed at an *arbitrary* random position in the arena. This is significantly more challenging than directly predicting the ground-truth factors, as the agent has to learn to reach the cube by acting on the joints, with a scalar reward as the only learning signal. Consequently, the compute required to learn this task is about 1,000 times greater than in the simple factor prediction case. We additionally include in our study a *pushing* task which consists of pushing an object to a goal position that is sampled at each episode. Learning this task takes one order of magnitude more compute than *object reaching*, likely due to the complex rigid-body dynamics and object interactions. To the best of our knowledge, this is the most challenging manipulation task that is currently feasible on our setup. Ahmed et al. (2021) report solving a similar pushing task, but require the full ground-truth state to be observable.

**Training the RL agents.** The inputs at time $t$ are the camera observation $o_t$ and a vector of observable variables $x_t$ containing the joint angles and velocities, as well as the target object position in *pushing*. We then feed the camera observation $o_t$ into an encoder $e$ that was pretrained on the dataset in Dittadi et al. (2021c). The result is concatenated with $x_t$, yielding a state vector $s_t = [x_t, e(o_t)]$. We then use SAC to train the policy with $s_t$ as input. The policy, value, and Q networks are implemented as MLPs with 2 hidden layers of size 256. When training the policies, we keep the encoder frozen.

**Model sweep.** To shed light on the research questions outlined in the previous sections, we perform a large-scale study in which we train 240 representation models and 11,520 downstream policies, as described below. See Appendix A for further implementation details.

- We train 120 $\beta$-VAEs (Higgins et al., 2017a) and 120 Ada-GVAEs (Locatello et al., 2020) with a subset of the hyperparameter configurations and neural architecture from Dittadi et al. (2021c). Specifically, we consider $\beta \in \{1, 2, 4\}$, $\beta$ annealing over $\{0, 50000\}$ steps, with and without

input noise, and 10 random seeds per configuration. The latent space size is fixed to 10 following prior work (Kim & Mnih, 2018; Chen et al., 2018; Locatello et al., 2020; Träuble et al., 2021).

- For *object reaching*, we train 20 downstream policies (varying random seed) for each of the 240 VAEs. The resulting 4,800 policies are trained for 400k steps (approximately 2,400 episodes).
- Since *pushing* takes substantially longer to train, we limit the number of policies trained on this task: We choose a subset of 96 VAEs corresponding to only 4 seeds, and then use 10 seeds per representation. The resulting 960 policies are trained for 3M steps (about 9,000 episodes).
- Finally, for both tasks we also investigate the role of regularization on the policy. More specifically, we repeat the two training sweeps from above (5,760 policies), with the difference that now the policies are trained with L1 regularization on the first layer.

**Limitations of our study.** Although we aim to provide a sound and extensive empirical study, such studies are inevitably computationally demanding. Thus, we found it necessary to make certain design choices. For each of these choices, we attempted to follow common practice, in order to maintain our study as relevant, general, and useful as possible. One such decision is that of focusing on autoencoder-based representations. To answer our questions on the effect of upstream representations on the generalization of downstream policies, we need a diverse range of representations. How these representations are obtained is not directly relevant to answer our research question. Following Dittadi et al. (2021c), we chose to focus on $\beta$-VAE and Ada-GVAE models, as they were shown to provide a broad set of representations, including fully disentangled ones. Although we conjecture that other classes of representation learning algorithms should generally reveal similar trends as those found in our study, this is undoubtedly an interesting extension. As for the RL algorithm used in this work, SAC is known to be a particularly sample-efficient model-free RL method that is a popular choice in robotics (Haarnoja et al., 2018a; Kiran et al., 2021; Singh et al., 2019). Extensive results on pushing from ground-truth features on the same setup in Ahmed et al. (2021) indicate that methods like TD3 (Fujimoto et al., 2018) or PPO (Schulman et al., 2017) perform very similarly to SAC under the same reward structure and observation space. Thus, we expect the results of our study to hold beyond SAC. Another interesting direction is the study of additional regularization schemes on the policy network, an aspect that is often overlooked in RL. We expect the potential insights from extending the study along these axes to not justify the additional compute costs and corresponding carbon footprint. However, with improving efficiency and decreasing costs, we believe these could become worthwhile validation experiments in the future.

## 4   RESULTS

We discuss our results in three parts: In Section 4.1, we present the training results of our large-scale sweep, and how policy regularization and different properties of the pretrained representations affect in-distribution reward. Section 4.2 gives an extensive account of which metrics of the pretrained representations predict OOD generalization of the agents in simulated environments. Finally, in Section 4.3 we perform a similar evaluation on the real robot, in a zero-shot sim-to-real scenario.

### 4.1   RESULTS IN THE TRAINING ENVIRONMENT

Fig. 2 shows the training curves of all policies for *object reaching* and *pushing* in terms of the task-specific success metric. Here we use success metrics for interpretability, as their range is always $[0, 1]$. In *object reaching*, the success metric indicates progress from the initial end effector position to the optimal distance from the center of the cube. It is 0 if the final distance is not smaller than the initial distance, and 1 if the end effector is touching the center of a face of the cube. In *pushing*, the success metric is defined as the volumetric overlap of the cube with the goal cube, and the task can be visually considered solved with a score around 80%.

From the training curves we can conclude that both tasks can be consistently solved from pixels using pretrained representations. In particular, all policies on *object reaching* attain almost perfect scores. Unsurprisingly, the more complex *pushing* task requires significantly more training, and the variance across policies is larger. Nonetheless, almost all policies learn to solve the task satisfactorily.

To investigate the effect of representations on the training reward, we now compute its Spearman rank correlations with various supervised and unsupervised metrics of the representations (Fig. 2 bottom). By training reward, we mean the average reward of a fully trained policy over 200 episodes

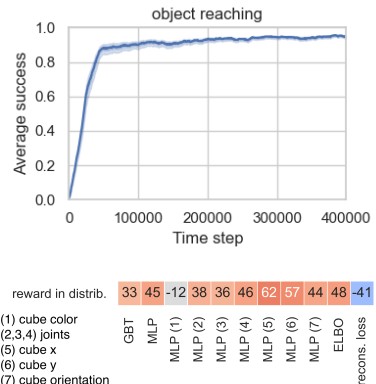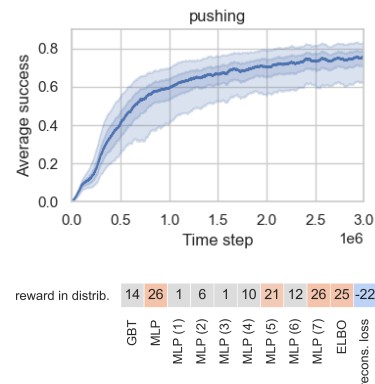

Figure 2: Top: Average training success, aggregated over *all* policies from the sweep (median, quartiles, 5th/95th percentiles). Bottom: Rank correlations between representation metrics and in-distribution reward (evaluated when the policies are fully trained), in the case without regularization. Correlations are color-coded in red (positive) or blue (negative) when statistically significant (p<0.05), otherwise they are gray.

in the training environment (see Appendix A). On *object reaching*, the final reward correlates with the ELBO and the reconstruction loss. A simple supervised metric to evaluate a representation is how well a small downstream model can predict the ground-truth factors of variation. Following Dittadi et al. (2021c), we use the MLP10000 and GBT10000 metrics (simply MLP and GBT in the following), where MLPs and Gradient Boosted Trees (GBTs) are trained to predict the FoVs from 10,000 samples. The training reward correlates with these metrics as well, especially with the MLP accuracy. This is not entirely surprising: if an MLP can predict the FoVs from the representations, our policies using the same architecture could in principle retrieve the FoVs relevant for the task. Interestingly, the correlation with the overall MLP metric mostly stems from the cube pose FoVs, i.e. those that are not included in the ground-truth state $x_t$. These results suggest that these metrics can be used to select good representations for downstream RL. On the more challenging task of *pushing*, the correlations are milder but most of them are still statistically significant.

**Summary.** Both tasks can be consistently solved from pixels using pretrained representations. Unsupervised (ELBO, reconstruction loss) and supervised (ground-truth factor prediction) in-distribution metrics of the representations are correlated with reward in the training environment.

## 4.2 OUT-OF-DISTRIBUTION GENERALIZATION IN SIMULATION

**In- and out-of-distribution rewards.** After training, the in-distribution reward correlates with OOD1 performance on both tasks (especially with regularization), but not with OOD2 performance (see Fig. 3). Moreover, rewards in OOD1 and OOD2 environments are moderately correlated across tasks and regularization settings.

**Unsupervised metrics and informativeness.** In Fig. 4 (left) we assess the relation between OOD reward and in-distribution metrics (ELBO, reconstruction loss, MLP, and GBT). Both ELBO and reconstruction loss exhibit a correlation with OOD1 reward, but not with OOD2 reward. These unsupervised metrics can thus be useful for selecting representations that will lead to more robust downstream RL tasks, as long as the encoder is in-distribution. While the GBT score is not correlated with reward under distribution shift, we observe a significant correlation between OOD1 reward and the MLP score,

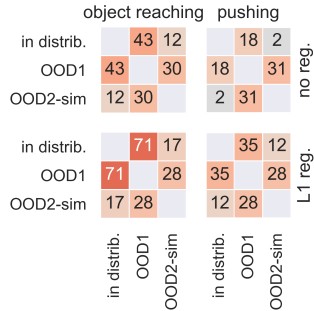

Figure 3: Correlations between training (in distrib.) and OOD rewards (p<0.05).

which measures downstream factor prediction accuracy of an MLP with the same architecture as the one parameterizing the policies. As in Section 4.1, we further investigate the source of this correlation, and find it in the pose parameters of the cube. Correlations in the OOD2 setting are much weaker, thus we conclude that these metrics do not appear helpful for model selection in this case. Our results on *pushing* confirm these conclusions although correlations are generally weaker, presumably due to the more complicated nature of this task. An extensive discussion is provided in Appendix B.2.

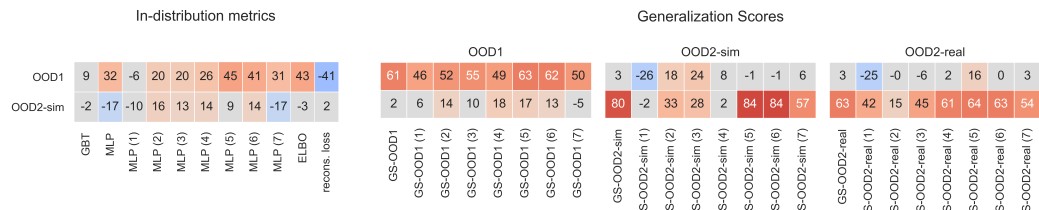

Figure 4: Rank correlations of representation properties with OOD1 and OOD2 reward on *object reaching* without regularization. Numbering when splitting metrics by FoV: (1) cube color; (2–4) joint angles; (5–7) cube position and rotation. Correlations are color-coded as described in Fig. 2.

**Correlations with generalization scores.** Here we analyze the link between generalization in RL and the generalization scores (GS) discussed in Section 2, which measure the generalization of downstream FoV predictors *out of distribution*, as opposed to the MLP and GBT metrics considered above. For both OOD scenarios, the distribution shifts underlying these GS scores are the same as the ones in the RL tasks in simulation. We summarize our findings in Fig. 4 (right) on the *object reaching* task. Reward in the OOD1 setting is significantly correlated with the GS-OOD1 metric of the pretrained representation. We observe an even stronger correlation between the reward in the simulated OOD2 setting and the corresponding GS-OOD2-sim and GS-OOD2-real scores. On a per-factor level, we see that the source of the observed correlations primarily stems from the generalization scores w.r.t. the pose parameters of the cube. The OOD generalization metrics can therefore be used as proxies for the corresponding form of generalization in downstream RL tasks. This has practical implications for the training of RL downstream policies which are generally known to be brittle to distribution shifts, as we can measure a representation's generalization score from a few labeled images. This allows for selecting representations that yield more robust downstream policies.

**Disentangled representations.** Disentanglement has been shown to be helpful for downstream performance and OOD1 generalization even with MLPs (Dittadi et al., 2021c). However, in *object reaching*, we only observe a weak correlation with some disentanglement metrics (Fig. 5). In agreement with (Dittadi et al., 2021c), disentanglement does not correlate with OOD2 generalization. The same study observed that disentanglement correlates with the informativeness of a representation. To understand if these weak correlations originate from this common confounder, we investigate whether they persist after adjusting for MLP FoV prediction accuracy. Given two representations with similar MLP accuracy, does the more disentangled one exhibit better OOD1 generalization? To measure this we predict success from the MLP accuracy using kNN (k=5) (Locatello et al., 2019) and compute the residual reward by subtracting the amount of reward explained by the MLP metric. Fig. 5 shows that this resolves the remaining correlations with disentanglement. Thus, for the RL downstream tasks considered here, disentanglement per se does not seem to be useful for OOD generalization. We present similar results on *pushing* in Appendix B.2.

**Policy regularization and observation noise.** It might seem unsurprising that disentanglement is not useful for generalization in RL, as MLP policies do not have any explicit inductive bias to exploit it. Thus, we attempt to introduce such inductive bias by repeating all experiments with L1 regularization on the first layer of the policy. Although regularization improves OOD1 and OOD2 generalization in general (see box plots in Fig. 5), we observe no clear link with disentanglement. Furthermore, in accordance with Dittadi et al. (2021c), we find that observation noise when training representations is beneficial for OOD2 generalization. See Appendix B.2 for a detailed discussion.

**Stronger OOD shifts: evaluating on a novel shape.** On *object reaching*, we also test generalization w.r.t. a novel shape by replacing the cube with a sphere. This corresponds to a strong OOD2-type shift, since shape was never varied when training the representations. Surprisingly, the policies appear to be robust to the novel shape. In fact, when the sphere has the same colors that the cube had during policy training, *all* policies get closer than 5 cm to the sphere on average, with a mean success metric of 95%. On sphere colors from the OOD1 split, more than 98.5% move the finger closer than this threshold, and on the strongest distribution shift (OOD2-sim colors, and cube replaced by sphere) almost 70% surpass that threshold with an average success metric above 80%.

**Summary.** (1) In- and out-of-distribution rewards are correlated, as long as the representation remains in its training distribution (OOD1). (2) Similarly, in-distribution representation metrics (both unsupervised and supervised) predict OOD1 reward, but are not reliable when the representation is

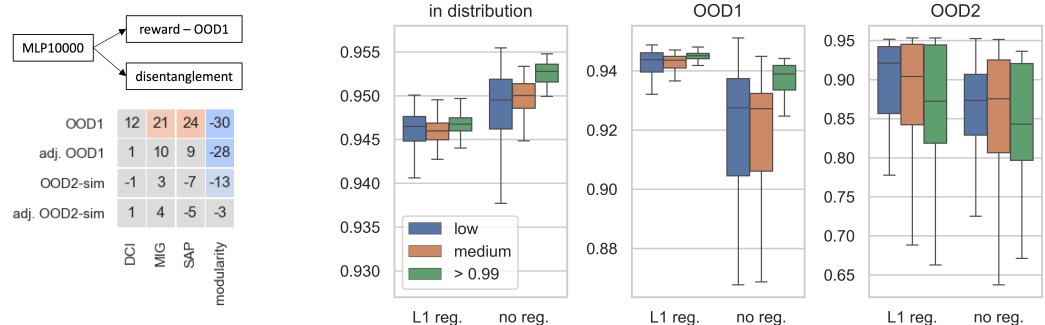

Figure 5: **Box plots**: fractional success on *object reaching* split according to low (blue), medium-high (orange), and almost perfect (green) disentanglement. L1 regularization in the first layer of the MLP policy has a positive effect on OOD1 and OOD2 generalization with minimal sacrifice in terms of training reward (see scale). **Correlation matrix** (left): although we observe a mild correlation between some disentanglement metrics and OOD1 (but not OOD2) generalization, this does not hold when adjusting for representation informativeness. Correlations are color-coded as described in Fig. 2. We use disentanglement metrics from Eastwood & Williams (2018); Chen et al. (2018); Kumar et al. (2018); Ridgeway & Mozer (2018).

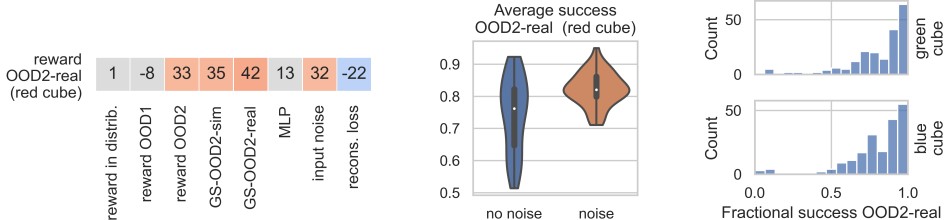

Figure 6: **Zero-shot sim-to-real** on *object reaching* on over 2,000 episodes. **Left:** Rank-correlations on the real platform with a red cube (color-coded as described in Fig. 2). **Middle**: Training encoders with additive noise improves sim-to-real generalization. **Right**: Histogram of fractional success in the more challenging OOD2-real-{green,blue} scenario from 50 policies across 4 different goal positions.

OOD (OOD2). (3) Disentanglement does not correlate with generalization in our experiments, while (4) input noise when training representations is beneficial for OOD2 generalization. (5) Most notably, the GS metrics, which measure generalization under distribution shifts, are significantly correlated with RL performance under similar distribution shifts. We thus recommend using these convenient proxy metrics for selecting representations that will yield robust downstream policies.

## 4.3 DEPLOYING POLICIES TO THE REAL WORLD

We now evaluate a large subset of the agents on the real robot without fine-tuning, quantify their zero-shot sim-to-real generalization, and find metrics that correlate with real-world performance.

**Reaching.** We choose 960 policies trained in simulation, based on 96 representations and 10 random seeds, and evaluate them on two (randomly chosen, but far apart) goal positions using a red cube. While a red cube was in the training distribution, we consider this to be OOD2 because real-world images represent a strong distribution shift for the encoder (Dittadi et al., 2021c; Djolonga et al., 2020). Although sim-to-real in robotics is considered to be very challenging without domain randomization or fine-tuning (Tobin et al., 2017; Finn et al., 2017; Rusu et al., 2017), many of our policies obtain a high fractional success without resorting to these methods. In addition, in Fig. 6 (left) we observe significant correlations between zero-shot real-world performance and some of the previously discussed metrics. First, there is a positive correlation with the OOD2-sim reward: Policies that generalize to unseen cube colors in simulation also generalize to the real world. Second, representations with high GS-OOD2-sim and (especially) GS-OOD2-real scores are promising candidates for sim-to-real transfer. Third, if no labels are available, the weaker correlation with the reconstruction loss on the simulated images can be exploited for representation selection. Finally, as observed by Dittadi et al. (2021c)

for simple downstream tasks, input noise while learning representations is beneficial for sim-to-real generalization (Fig. 6, middle).

Based on these findings, we select 50 policies with a high GS-OOD2-real score, and evaluate them on the real world with a green and a blue cube, which is an even stronger OOD2 distribution shift. In Fig. 6 (right), where metrics are averaged over 4 cube positions per policy, we observe that most policies can still solve the task: approximately 80% of them position the finger less than 5 cm from the cube. Lastly, we repeat the evaluations on the green sphere that we previously performed in simulation, and observe that many policies successfully reach this completely novel object. See Appendix B.3 and the project website for additional results and videos of deployed policies.

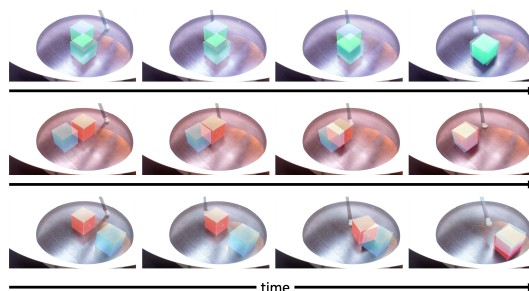

Figure 7: We select pushing policies with high GS-OOD2-real score. When deployed on the real robot without fine-tuning, they succeed in pushing the cube to a specified goal position (transparent blue cube).

**Pushing.** We now test whether our real-world findings on *object reaching* also hold for *pushing*. We again select policies with a high GS-OOD2-real score and encoders trained with input noise. We record episodes on diverse goal positions and cube colors to support our finding that pushing policies in simulation can generalize to the real robot. In Fig. 7, we show three representative episodes with successful task completions and refer to the project site for video recordings and further episodes.

**Summary.** Policies trained in simulation can solve the task on the real robot without domain randomization or fine-tuning. Reconstruction loss, encoder robustness, and OOD2 reward in simulation are all good predictors of real-world performance. For real-world applications, we recommend using GS-OOD2-sim or GS-OOD2-real for model selection, and training the encoder with noise.

## 5    OTHER RELATED WORK

A key unsolved challenge in RL is the brittleness of agents to distribution shifts in the environment, even if the underlying structure is largely unchanged (Cobbe et al., 2019; Ahmed et al., 2021). This is related to studies on representation learning and generalization in downstream tasks (Gondal et al., 2019; Steenbrugge et al., 2018; Dittadi et al., 2021b; Esmaeili et al., 2019; Chaabouni et al., 2020), as well as domain generalization (see Wang et al. (2021) for an overview). More specifically for RL, Higgins et al. (2017b) focus on domain adaptation and zero-shot transfer in DeepMind Lab and MuJoCo environments, and claim disentanglement improves robustness. To obtain better transfer capabilities, Asadi et al. (2020) argue for discretizing the state space in continuous control domains by clustering states where the optimal policy is similar. Kulkarni et al. (2015) propose geometric object representations by means of keypoints or image-space coordinates and Wulfmeier et al. (2021) investigate the effect of different representations on the learning and exploration of different robotics tasks. Transfer becomes especially challenging from the simulation to the real world, a phenomenon often referred to as the sim-to-real gap. This is particularly crucial in RL, as real-world training is expensive, requires sample-efficient methods, and is sometimes unfeasible if the reward structure requires accurate ground truth labels (Dulac-Arnold et al., 2019; Kormushev et al., 2013). This issue is typically tackled with large-scale domain randomization in simulation (Akkaya et al., 2019; James et al., 2019).

## 6    CONCLUSION

Robust out-of-distribution (OOD) generalization is still one of the key open challenges in machine learning. We attempted to answer central questions on the generalization of reinforcement learning agents in a robotics context, and how this is affected by pretrained representations. We presented a large-scale empirical study in which we trained over 10,000 downstream agents given pretrained representations, and extensively tested them under a variety of distribution shifts, including sim-to-real. We observed agents that generalize OOD, and found that some properties of the pretrained representations can be useful to predict which agents will generalize better. We believe this work brings us one step closer to understanding the generalization abilities of learning systems, and we hope that it encourages many further important studies in this direction.

## ETHICS STATEMENT

Our study is based on synthetic and real data of a robotic setup where a robotic finger interacts with a cube. Our study does therefore not involve any human subjects leading to discrimination, bias or fairness concerns, or privacy and security issues. Representation learning and generalization is important across many disciplines and applications and could have harmful consequences without humans in the loop in safety-relevant settings. Having a sound understanding of the robustness of a given ML system based on such pretrained representations to distributions shifts is crucial to avoid harmful consequences in potential future high-stake applications to society, such as human-robot interaction (e.g. robotic surgery), autonomous driving, healthcare applications or other fairness-related settings. Here, we investigate a narrow aspect of this, that is, learning arguably harmless manipulation skills like reaching or pushing an object with a simple robotic finger. Our conclusions for OOD generalization are based on this setting and thus cannot be directly transferred to any given application setting of concern.

## REPRODUCIBILITY STATEMENT

To make sure our experiments are fully reproducible, we provided a full account of all required setup and implementation details in Section 3 and Appendix A.

## ACKNOWLEDGEMENTS

We would like to thank Anirudh Goyal, Georg Martius, Nasim Rahaman, Vaibhav Agrawal, Max Horn, and the Causality group at the MPI for useful discussions and feedback. We thank the International Max Planck Research School for Intelligent Systems (IMPRS-IS) for supporting FT. Part of the experiments were generously supported with compute credits by Amazon Web Services.

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

## A  IMPLEMENTATION DETAILS

**Task definitions and reward structure.**    We derive both tasks, *object reaching* and *pushing*, from the CausalWorld environments introduced by Ahmed et al. (2021). We pretrain representations on the dataset introduced by Dittadi et al. (2021c), and allow only one finger to move in our RL experiments. We introduce the *object reaching* environment that involves an unmovable cube. We used reward structures similar to those in Ahmed et al. (2021):

- *object reaching*: $r_t = -750 \left[ d(g_t, e_t) - d(g_{t-1}, e_{t-1}) \right]$
- *pushing*: $r_t = -750 \left[ d(o_t, e_t) - d(o_{t-1}, e_{t-1}) \right] - 250 \left[ d(o_t, g_t) - d(o_{t-1}, g_{t-1}) \right] + \rho_t$

where $t$ denotes the time step, $\rho_t \in [0, 1]$ is the fractional overlap with the goal cube at time $t$, $e_t \in \mathbf{R}^3$ is the end-effector position, $o_t \in \mathbf{R}^3$ the cube position, $g_t \in \mathbf{R}^3$ the goal position, and $d(\cdot, \cdot)$ denotes the Euclidean distance. The cube in *object reaching* is fixed, i.e. $o_t = g_t$ for all $t$. The time limit is 2 seconds in *object reaching* and 4 seconds in *pushing*.

**Success metrics.**    Besides the accumulated reward along episodes, that is determined by the reward function, we also report two reward-independent normalized success definitions for better interpretability: In *object reaching*, the success metric indicates progress from the initial end effector position to the optimal distance from the center of the cube. It is 0 if the final distance is greater than or equal to the initial distance, and 1 if the end effector is touching the center of a face of the cube. In *pushing*, the success metric is defined as the volumetric overlap of the cube with the goal cube, and the task can be visually considered solved with a score around 80%. We observed that accumulated reward and success are very strongly correlated, thus allowing us to use one or the other for measuring performance.

**Training and evaluation details.**    During training, the goal position is randomly sampled at every episode. Similarly, the object color is sampled from the 4 specified training colors from $\mathcal{D}_1$ that correspond to the OOD1-B split from Dittadi et al. (2021c).

For each policy evaluation (in-distribution and out-of-distribution variants), we average the accumulated reward and final success over 200 episodes with randomly sampled cube positions and the respective object color in both tasks.

**SAC implementation.**    Our implementation of SAC builds upon the `stable-baselines` package (Hill et al., 2018). We use the same SAC hyperparameters used for pushing in Ahmed et al. (2021). When using L1 regularization, we add to the loss function the L1 norm of the first layers of all MLPs, scaled by a coefficient $\alpha$. We gradually increase regularization by linearly annealing $\alpha$ from 0 to $5 \cdot 10^{-7}$ over 200,000 time steps in *object reaching*, and from 0 to $6 \cdot 10^{-8}$ over 3,000,000 time steps in *pushing*.

## B  ADDITIONAL RESULTS

### B.1  TRAINING ENVIRONMENT

Fig. 2 in Section 4.1 shows correlations of unsupervised and supervised metrics with in-distribution reward for *object reaching* and *pushing*, only in the case without regularization. In Fig. 8 we also show these results in the case with regularization, as well as when adjusting for MLP informativeness.

### B.2  OUT-OF-DISTRIBUTION GENERALIZATION IN SIMULATION

In Section 4.2 we discussed rank-correlations of OOD rewards with unsupervised, informativeness and generalization scores on *object reaching* without regularization. In Fig. 9 we also show these results for the case with regularization and on *pushing*, as well as when adjusting for MLP informativeness. Without regularization, we observe on *pushing* very similar correlations along all metrics as we observed on *object reaching*, confirming our conclusions on this more complex task. When using regularization, rank correlations are very similar across both tasks. Interestingly, the correlation

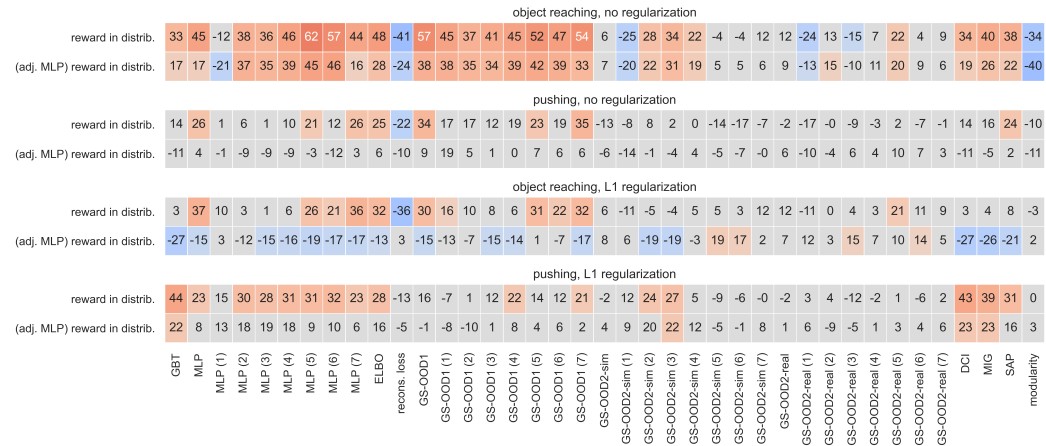

Figure 8: Rank correlations between metrics and in-distribution reward, with and without adjusting for informativeness. Correlations are color-coded as described in Fig. 2.

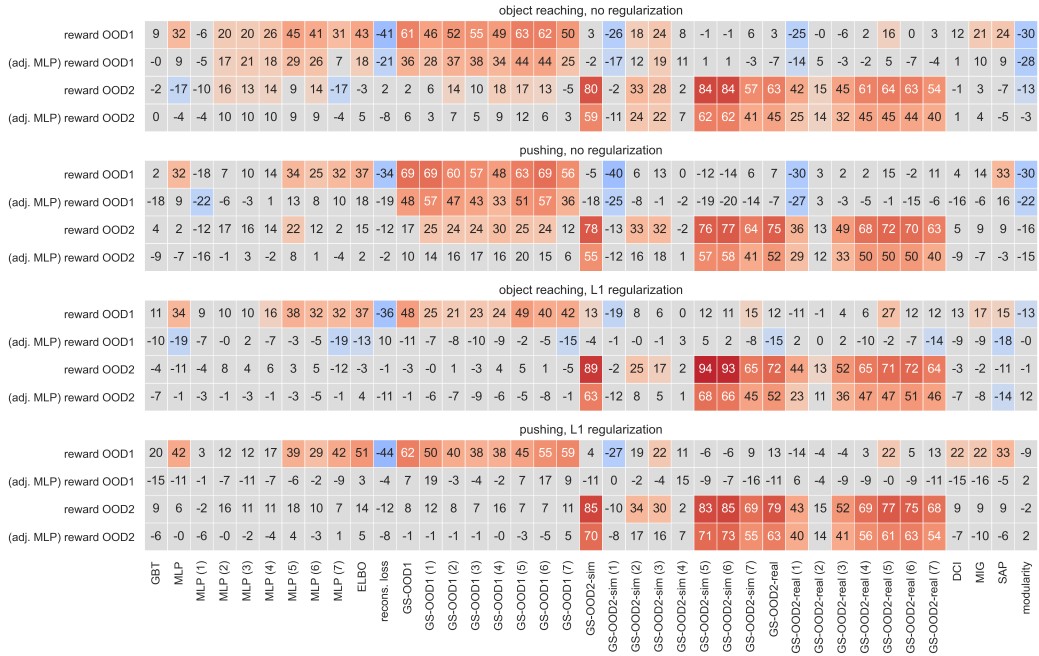

Figure 9: Rank correlations between metrics and OOD reward, with and without adjusting for informativeness. Correlations are color-coded as described in Fig. 2.

between GS-OOD2 scores and OOD2 generalization of the policy is even stronger when using L1 regularization. In contrast to our observations without regularization, we find that the correlation between GS-OOD1 and OOD1 generalization of the policy vanishes when adjusting for the MLP metric.

### B.2.1 DISENTANGLED REPRESENTATIONS

As discussed in Section 4.2 for *object reaching* without regularization, we observe in Fig. 9 a weak correlation between some disentanglement metrics and OOD1 reward, which however vanishes when adjusting for MLP informativeness. In agreement with Dittadi et al. (2021c), we observe no significant correlation between disentanglement and OOD2 generalization, for both tasks, with and without regularization. From Fig. 10 we see that in some cases, especially without regularization, a very high DCI score seems to lead to better performance on average. However, this behavior is

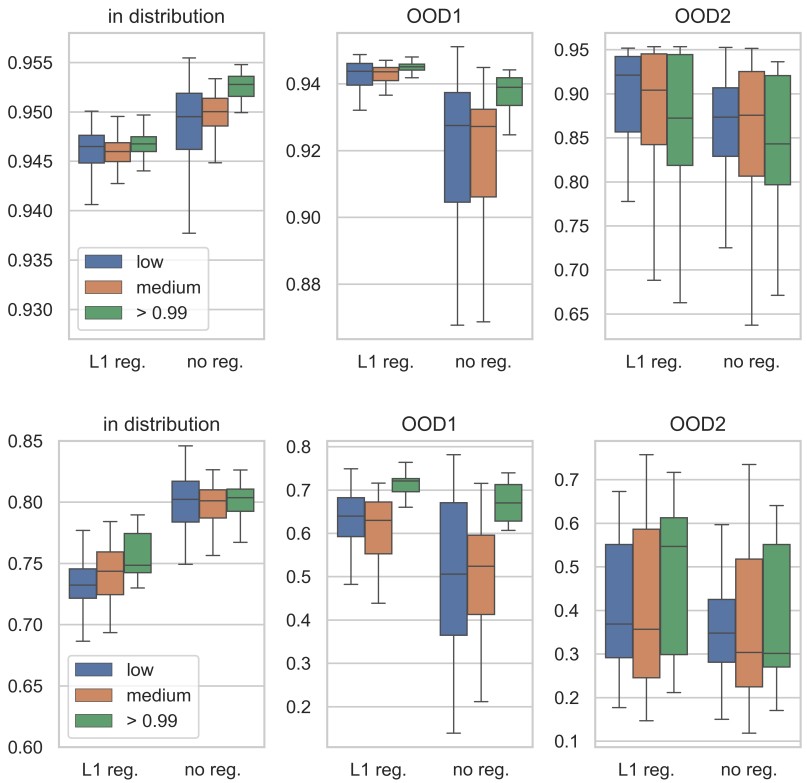

Figure 10: Fractional success on *object reaching* (**top**) and *pushing* (**bottom**), split according to low (blue), medium-high (orange), and almost perfect (green) disentanglement. Results for *object reaching* are also reported in Fig. 5 in Section 4.2.

not significant (within error bars), as opposed to the results shown in simpler downstream tasks by Dittadi et al. (2021c). Furthermore, this trend is likely due to representation informativeness, since the correlations with disentanglement disappear when adjusting for the MLP score, as discussed above.

### B.2.2 REGULARIZATION

As seen in Fig. 10, regularization generally has a positive effect on OOD1 and OOD2 generalization, which is particularly prominent in the OOD1 setting. On the other hand, it leads to lower training rewards both in *object reaching* and in *pushing*. In the latter, the performance drop is particularly significant, while in *object reaching* it is negligible.

### B.2.3 SAMPLE EFFICIENCY

In addition to the analysis reported in the main paper, we investigate how representation properties affect sample efficiency. Specifically, we store checkpoints of our policies at $t \in \{20\text{k}, 50\text{k}, 100\text{k}, 400\text{k}\}$ for *object reaching* and $t \in \{200\text{k}, 500\text{k}, 1\text{M}, 3\text{M}\}$ for *pushing*. We then evaluate policies at these time step on the same three environments as before: (1) on the cube colors from training; (2) on the OOD1 cube colors; and (3) on the OOD2-sim cube colors. Results are summarized in Fig. 11 for *object reaching* and Fig. 12 for *pushing*.

On *object reaching* (Fig. 11), we observe very similar trends with and without regularization: Unsupervised metrics (ELBO and reconstruction loss) display a correlation with the training reward, as do the supervised informativeness metrics (GBT and MLP). This is strongest on early timesteps, meaning these scores could be important for sample efficiency. Similarly, we observe a correlation with the disentanglement scores DCI, MIG and SAP. With the help of the additional evaluation of

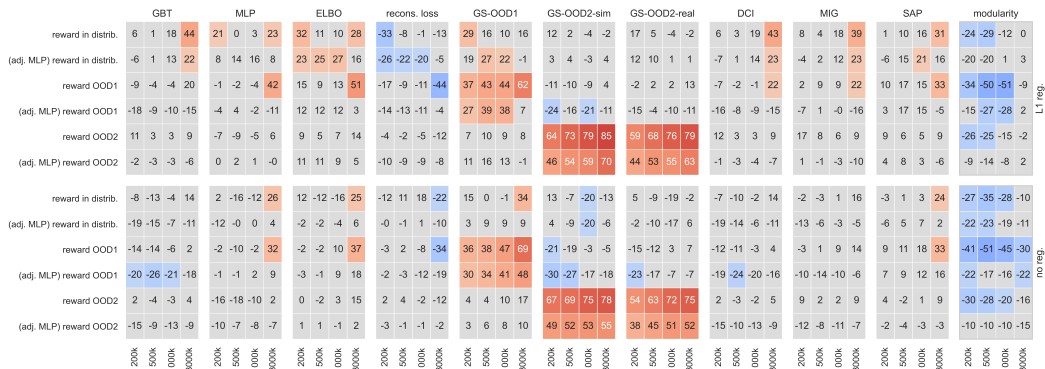

Figure 11: Sample efficiency analysis for *object reaching*. Rank correlations of rewards with relevant metrics along multiple time steps. Correlations are color-coded as described in Fig. 2.

Figure 12: Sample efficiency analysis for *pushing*. Rank correlations of rewards with relevant metrics along multiple time steps. Correlations are color-coded as described in Fig. 2.

rewards adjusted for MLP informativeness, we can attribute this correlation again to this common confounder. Crucially, we see that the generalization scores (GS) are correlated with generalization of the corresponding policies under OOD1 and OOD2 shifts for all recorded time steps, confirming the results in the main text.

On *pushing* (Fig. 12), many correlations at early checkpoints are significantly reduced, especially with regularization. This behavior might be due to the more complicated nature of the task, which involves learning to reach the cube first, and then push it to the goal. Correlations are primarily seen towards the end of training, with similar spurious correlations with disentanglement as elaborated above. Importantly, correlations between generalization scores (GS) and policy generalization under the same distribution shifts remain strong and statistically significant, corroborating the analysis in the main text.

### B.2.4 GENERALIZATION TO A NOVEL SHAPE

As mentioned in Section 4.2, on the *object reaching* task, we also test generalization w.r.t. a novel object shape by replacing the cube with an unmovable sphere. This corresponds to a strong OOD2-type shift, since shape was never varied when training the representations. We then evaluate a subset of 960 trained policies as before, with the same color splits. Surprisingly, the policies appear to handle the novel shape as we see from the histograms in Fig. 13 in terms of success and final distance. In fact, when the sphere has the same colors that the cube had during policy training, *all* policies get closer than 5 cm to the sphere on average, with a mean success metric of about 95%. On sphere colors from the OOD1 split, more than 98.5% move the finger closer than this threshold, and on the strongest distribution shift (OOD2-sim colors and cube replaced by sphere) almost 70% surpass that threshold with an average success metric above 80%.

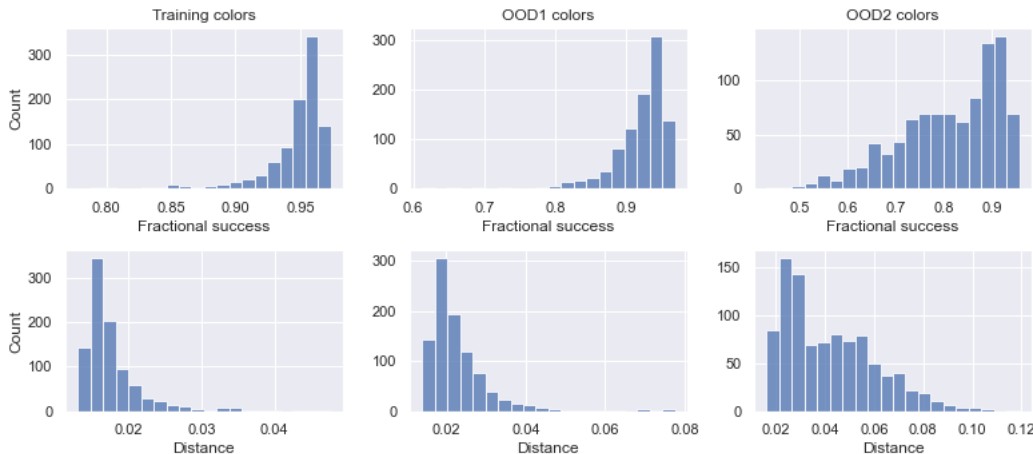

Figure 13: Testing policies for *object reaching* under the same in-distribution, OOD1, and OOD2 evaluation protocols regarding object color in simulation, but replacing the cube with a sphere, which was never used in training.

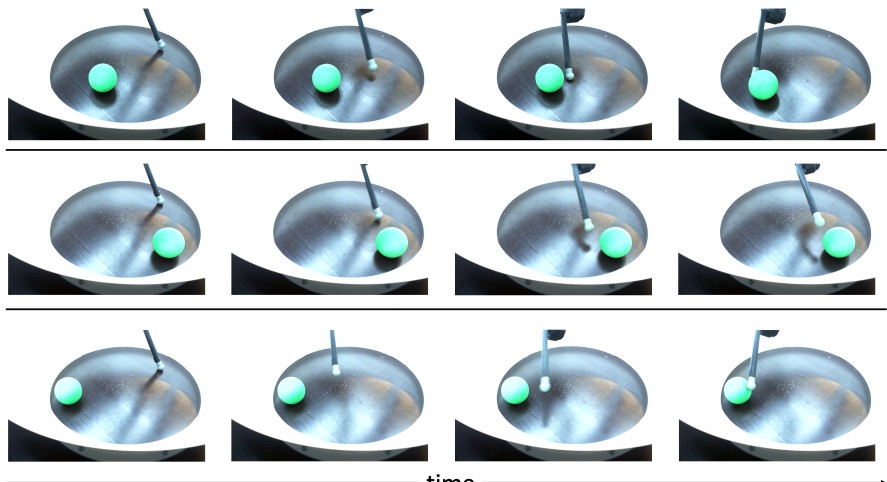

Figure 14: Transferring policies for *object reaching* to the real robot setup without any fine-tuning on a green sphere (unseen shape *and* color). Correlations are color-coded as described in Fig. 2.

## B.3    DEPLOYING POLICIES TO THE REAL WORLD

In Fig. 14 we show three representative episodes of testing a reaching policy on the real robot for the strong OOD shift with a novel sphere object shape instead of the cube from training. We present the respective videos in the project page. There we also present videos of additional real-world episodes on pushing and reaching cubes of different colors.

