# OpenReview forum: "The Role of Pretrained Representations for the OOD Generalization of RL Agents"
_ICLR.cc/2022/Conference — ICLR 2022 Poster_

### Official Review · Reviewer_YvcV · 2021-10-31

**Correctness:** 4
**Technical Novelty And Significance:** 2
**Empirical Novelty And Significance:** 2
**Recommendation:** 5
**Confidence:** 4

**Main Review:**

Strengths:

* The paper studies the effect of pretraining visual representation on the downstream out-of-distribution task performance.
* The paper conducted a relatively thorough analysis on this issue. More specifically, the paper studies the effect of VAE pretraining on the TriFinger environment. The studied factors include the measuring metric, network regularization, input noise, different OOD distributions.
* While most of the conclusions are intuitive to understand, it's good to see that the paper has many experiments to support them.

Weaknesses:

* While the paper did a thorough analysis on VAE and in the TriFinger environment, I found these two are limiting. First, there are many different pretraining techniques such as contrastive learning and using auxilliary loss. This paper only studies VAE. Therefore, the paper title is a bit over-claimed. Second, TriFinger is a challenging control task but not necessarily a difficult task in terms of perception. As we can see in the example environment figures (Figure 1), the image input to the policy is relatively simple and has a clean background. Therefore, it is possible that the perception network can easily learn to only focus on the object in the image and ignore other things during training. And such a network can be more robust than expected and can transfer to the real world better. Therefore, the conclusions drawn in this paper only apply to VAE pre-training and TriFinger environments. It remains unclear whether the conclusions still hold in visually more complex environments such as ProcGen environments.

* Even just in the TriFinger environments, a more systematic analysis should include the testings on the TriFinger environments with different backgrounds. As we can see in Figure 1, the bowl environments are very clean and look similar in simulation and in the real world. Since the pretraining is mainly about training the perception module, OOD tests should also change the bowl texture, color during the testing.


**Summary Of The Paper:**

This paper is an analysis paper that uses well-known existing pretraining techniques to study the effect of pretrained representations on out-of-distribution generalization performance. The paper mainly uses the VAEs to train the visual representation network. While I believe this is an important topic, the paper only analyzed one pretraining technique and in a visually simple environment.

**Summary Of The Review:**

I believe the paper is analyzing an important issue in reinforcement learning. However, more experiments on TriFinger with different visual backgrounds and other visually complex environments such as Procgen, and experiments with different pretraining techniques are needed to make the analysis more thorough and convincing.

---

> ### Author Response · Authors · 2021-11-15
> **Response to Reviewer YvcV**
>
> We thank the reviewer for their time and effort, and are encouraged to see that the reviewer appreciates the relevance of the topic and the breadth of experiments to support our conclusions. We address the reviewer’s feedback below and hope they will consider adjusting their assessment of our work based on these clarifications.
>
> &nbsp;
>
> **Using only VAEs.**
>
> - This choice is motivated in the “Limitations of our study” paragraph (Section 3), which was appreciated by other reviewers. The main point is that how the representations are obtained is not directly relevant to answer our research questions. Following Dittadi et al. (2021), we chose β-VAE and Ada-GVAE and swept over some of their hyperparameters, to get a diverse set of representations, including fully disentangled ones.
>
> - On the potential overclaiming issue in the title: We accurately explain the scope of the paper including the usage of VAEs for this analysis in the abstract. We are open to suggestions regarding the title.
>
> **TriFinger complexity.**
>
> - **“The perception network can easily learn to only focus on the object”**: The encoder actually focuses on *all* the factors of variation, since (1) these can be predicted by a downstream classifier with high accuracy, and (2) the VAEs accurately reconstruct the input images, which would not be possible if the factors not related to the object were not inferred correctly. We are happy to clarify this and include further quantitative results to support it.
>
> - **“Conclusions drawn in this paper only apply to [...] TriFinger environments.”** As in all empirical studies, certain design decisions necessarily had to be made. We do not have access to other robotic environments and we would like to emphasize the amount of work and resources that are required in robotics to conduct such a large-scale analysis. It is common to base such studies in RL on one specific robotics platform, see e.g. [1,2,3,4].
>
> - **“unclear whether the conclusions still hold in visually more complex environments such as ProcGen.”** While considering more complex visual environments is an interesting direction, it would require corresponding real-world robotic platforms. As for the ProcGen example, this would not fit the scope of our paper since we focus on robotics.
>
> **Include the testings on the TriFinger environments with different backgrounds.**
> We intentionally used the same environment variations as Dittadi et al to keep it comparable. In addition, changing the background would be a strong (potentially unrealistic) distribution shift, as the agents always saw only a single background color during training. Most importantly, we wanted to study OOD distribution shifts that we can also evaluate in the real world and we are therefore tied to the visual appearance of the one real-world robotic platform we do have. We therefore cannot easily change background colors. However, we also think the object color and shape distribution shifts are much more relevant for the tasks considered here.
>
> &nbsp;
>
> **References**
>
> [1] Wulfmeier, Markus, et al. "Representation matters: Improving perception and exploration for robotics." ICRA 2021.
>
> [2] Lee et al. "Beyond Pick-and-Place: Tackling Robotic Stacking of Diverse Shapes." CoRL 2021.
>
> [3] Higgins, Irina, et al. "Darla: Improving zero-shot transfer in reinforcement learning." International Conference on Machine Learning. PMLR, 2017.
>
> [4] Andrychowicz, OpenAI: Marcin, et al. "Learning dexterous in-hand manipulation." The International Journal of Robotics Research 39.1 (2020): 3-20.

---

> > ### Comment · Reviewer_YvcV · 2021-11-24
> > **Response**
> >
> > I appreciate the authors' reply. Some of my concerns are addressed. I am happy to raise the score to 5.
> >
> > * I can now see why the authors only use VAE for the pre-training.
> >
> > * Regarding the TriFinger complexity, my point is that TriFinger is a visually simple environment. "The perception network can easily learn to only focus on the object" is just an example of what the network could be learning. As argued by the authors, it could be learning something else too. But this does not change the fact that TriFinger environments are visually simple and learning a good perception module, in this case, can be relatively easy.
> >
> > * While I can totally understand that training for many RL environments (_not just robotics_) takes a significant amount of time, but it is hard to see how conclusions drawn in this paper can be extended to other OOD situations. As Reviewer BEPU also agrees that _to some extent_, the domain shift in the four scenarios presented in the paper is not that significant. This also connects to the point on _why other setups that have more significant domain shifts are not considered_. For example, we can easily change the texture and color of the bowls , the size of the bowl and cubes, the object shapes, even the color of the arm, in _simulation_ (I am not asking for any real robotic experiments). Why are these OOD situations not considered? Changing the background color is definitely a realistic scenario. Just from a practical perspective, we would want our robots to work out of lab environments. In the real world, the bowl can of course has any kind of color. There is a difference between what real robot experiment conditions that the authors have access to and how the real-world scenarios actually are. I understand that the authors cannot change the bowl color used in their real-world experiment. But why not provide the evaluation in the simulation? Since the paper is all about OOD, it is important to carefully design the experiments and do a thorough analysis. The real-world experiments are a big plus, but should not be the reason why the authors did not do a thorough analysis in simulation (in simulation, many factors can be changed, and therefore the analysis can be more thorough).
> >
> > * The authors believe that changing the background color is a strong distribution shift. What is the criterion that authors use to define how strong a distribution shift is? For a new task, how can one tell what setups are small distribution shifts, what setups are strong shifts? And where is the distribution shift boundary within which one can possibly use the conclusions drawn in this paper? While some of these questions might be hard to answer, but the authors should describe them in the limitation section.
> >
> > * With the aforementioned reasons, I believe the paper needs more systematic analysis from the machine learning perspective. But I do agree that this paper is valuable from the robotics perspective. I suggest that this paper is more suitable for a robotics conference.

---

> > > ### Author Response · Authors · 2021-11-25
> > > **Thanks for the feedback**
> > >
> > > We are very grateful to the reviewer for engaging in the discussion and reconsidering the score. In case the reviewer hasn’t noticed it yet, we would like to point out the update at the end of vE69’s review.
> > >
> > > We reply to the reviewer’s remaining concerns below.
> > >
> > > &nbsp;
> > >
> > > **“TriFinger environments are visually simple and learning a good perception module, in this case, can be relatively easy.”** Although these environments are relatively simple visually (but only in simulation), we would like to point out that this is still far from trivial, as also discussed by Dittadi et al. (2021). Moreover, this is not a toy scenario: it corresponds to a real robotic platform, where policies trained in simulation are deployed. Therefore, we believe this does not limit the scope of our work. In any case, our focus is not on learning representations in a visually very complex environment, but rather understanding the role of representations for generalization in challenging downstream tasks.
> > >
> > > &nbsp;
> > >
> > > **About evaluating on more OOD scenarios.** We already consider 7 distribution shifts (2 different shifts of cube color in simulation, shape shift in simulation with the same 3 distinct sets of colors as in the cube experiments, sim-to-real, sim-to-real with novel shape). We believe that the additional shifts in simulation suggested by the reviewer should not provide new insights, because:
> > >
> > > - The shifts in our study range from relatively small shifts (e.g. only shape or only cube color) to stronger shifts affecting the entire image (typical sim-to-real scenario in robotics);
> > > - The small shifts are already clearly noticeable on the downstream task (Fig. 5, right);
> > > - We have already observed similar trends even between radically different shifts (e.g. the cube color in simulation and the sim-to-real shift);
> > > - To strengthen this point, we have now computed the rank correlations between evaluations in simulation with cube and sphere (these are not in the paper yet, but we will add them for completeness). We observed a correlation of 0.32 between the rewards (i) under a OOD2 color shift of the cube (OOD2-sim), and (ii) under a shape shift from cube to sphere. This is very similar to the correlation between OOD2-sim and real robot reward (third number in Fig. 6 left).
> > >
> > > Moreover, since we performed an extensive study with over 10k trained policies, even if we were to evaluate only on reaching, and only on one of the OOD experiments suggested by the reviewer (e.g. the bowl color experiment), we would still need to run almost 2 million episodes in total (e.g. approximately 25 days on 64 cores).
> > >
> > > In conclusion, we are not convinced by the arguments of the reviewer that additional OOD2 shifts are necessary. We are however still open to discussion, and we would gladly run additional evaluations, given reasonable arguments that new interesting insights can be expected.
> > >
> > > &nbsp;
> > >
> > > **Strength/effect of distribution shifts.** We agree that precisely characterizing distribution shifts a priori is a hard open problem, and to the best of our knowledge there is no way to reliably predict whether a different distribution shift will have more or less impact than those considered in this paper. However, as argued in the previous paragraph, since we observed similar trends in different distribution shifts, we believe that additional shifts in simulation are not worth the cost. We are happy to include a discussion on this in the limitations section.

---

### Official Review · Reviewer_BEPU · 2021-10-31

**Correctness:** 3
**Technical Novelty And Significance:** 1
**Empirical Novelty And Significance:** 2
**Recommendation:** 3
**Confidence:** 4

**Main Review:**

Strengths
- OoD generalization of RL agents is a very active area of research. The paper provides a systematic and thorough empirical study on the impact of pretrained representations.
- The correlation analysis between different proxy metrics and the OoD generalization performance is extensive and sound.

Weakness
- While the key claim of the paper lies in OoD generalization, a large part of experiments are essentially not OoD. Among the four scenarios (IID, OoD1, OoD2, and real-world), only the OoD2 presents a clear domain shift to the pre-trained encoder.
- It seems not very clear to which degree the *agents generalize OoD*. A more direct comparison between IID, OoD1 and OoD2 would perhaps make claim 1 about OoD generalization more convincing.
- The core of claim 2, i.e., the GS metric can be used for representation selection, looks over-stretched to me since:
  - it requires labeled samples from the OoD domain, which is often not realistic
  - it requires ground truth values of the underlying factors of variation, which is often not realistic neither
- Finally, the experiment design overall looks quite incremental to [A Dittadi et al. ICLR'21].

**Summary Of The Paper:**

The paper presents an extensive empirical evaluation of the VAE-based pretrained representations for OoD generalization of RL agents in a robotic setup.

From the reported experiment results, it draws two conclusions:
1) agents build on top of the pretrained representations can generalize to some challenging OoD scenarios;
2) the prediction error of the ground truth values of the underlying factors of variations can be used as a proxy metric for selecting representation for OoD agents.

**Summary Of The Review:**

While the empirical evaluation is extensive and rigorous, the two key claims made in the paper are debatable. I, therefore, recommend reject. I'd be happy to change my rating if the author can better support the two claims during rebuttal.

---

> ### Author Response · Authors · 2021-11-15
> **Response to Reviewer BEPU**
>
> We thank the reviewer for their time and effort, and are encouraged to see that the reviewer considers our empirical study to be “systematic and thorough” with a “correlation analysis between different proxy metrics and the OoD generalization performance [that] is extensive and sound.” We are concerned that there might be a misunderstanding regarding the nature of the distribution shifts in our study. We attempt to resolve this misunderstanding below, along with the rest of the reviewer’s comments. We would be grateful if the reviewer could consider re-evaluating our work in light of these clarifications.
>
> &nbsp;
>
> **Comments 1 and 2 on OOD evaluation.**
>
> 1. **“A large part of experiments are essentially not OoD.”** As explained in Section 2, OOD1, OOD2-sim and OOD2-real are all OOD scenarios. OOD1 and OOD2-sim have cube colors that are OOD w.r.t. the policy, and in particular the colors in OOD2-sim are OOD w.r.t. the encoder as well. Such OOD shifts with respect to underlying data generating factors are widely used in the literature of OOD generalization [Montero et al 2021, Gondal et al. 2019, Schott et al 2021, Dittadi et al 2021, Träuble et al 2021, Locatello et al 2020]. Finally, OOD2-real is OOD in the well-known sense of sim-to-real distribution shift.
>
> 2. **“More direct comparison between IID, OoD1 and OoD2'' to clarify to which degree the agents generalize OoD.** If the concern is about the evaluation scenarios and how much out-of-distribution they are, we refer to the paragraph above, to Section 2 in our paper, and to Dittadi et al. 2021. If the concern is about relative performance on these different evaluation scenarios, we refer to the box plots in Fig. 5, the middle and right plots in Fig. 6, as well as Figures 10 and 13 in Appendix, and the additional videos on the project website for qualitative results.
>
> **The claim that GS metrics can be used for representation selection is overstretched.**
> We respectfully disagree with this assessment:
>
> - First, we believe it is reasonable to assume that labels are available in simulation. In fact, simulators have access to these symbolic variables as they are necessary for generating the observations. Thus, evaluating in the simulated scenarios (in distribution, OOD1, OOD2-sim) should not be any problem.
>
> - Second, we show in Fig. 6 (left) that for the OOD2-real case the GS-OOD2-sim metric is effective: using the metric in simulation is sufficient, and collecting labels on the real world is not necessary.
>
> - In addition, we remark that evaluating the GS metrics only involves *evaluating* the downstream models, so we do not necessarily need as many labels as we would need for training. Therefore, even a few real-world labels can already be useful.
>
> - Finally, even in the real world, it is not necessarily unrealistic to have access to the labels of these factors. For example, the cube position could be obtained with an existing object tracking method.
>
> **Incremental experimental setup, similar to Dittadi et al.**
> We intentionally match the experimental setup and distribution shifts of Dittadi et al. as closely as possible, such that we can directly relate the results on their simple downstream task with those on our more challenging RL tasks. Sections 1, 2, and 3 of our paper motivate in detail why we use exactly the same setup and how the empirical study differs from Dittadi et al. In particular, this work significantly differs in the downstream tasks, which are robotics control tasks that are considerably more challenging than the factor prediction considered in Dittadi et al. A good reference for how hard the training and sim-to-real transfer of control policies is Lee et al. 2021.
>
> &nbsp;
>
> **References**
>
> Montero et al. The role of disentanglement in generalisation. ICLR 2021
>
> Gondal et al. On the transfer of inductive bias from simulation to the real world: a new disentanglement dataset. NeurIPS 2019.
>
> Schott et al. Visual representation learning does not generalize strongly within the same domain. arXiv preprint arXiv:2107.08221.
>
> Dittadi et al. On the transfer of disentangled representations in realistic settings. ICLR 2021.
>
> Träuble et al. On disentangled representations learned from correlated data. ICML 2021.
>
> Locatello et al. Weakly-supervised disentanglement without compromises. ICML 2020.
>
> Lee et al. "Beyond Pick-and-Place: Tackling Robotic Stacking of Diverse Shapes." CoRL 2021.

---

### Official Review · Reviewer_AwPM · 2021-11-02

**Correctness:** 4
**Technical Novelty And Significance:** 2
**Empirical Novelty And Significance:** 3
**Recommendation:** 6
**Confidence:** 4

**Main Review:**

The paper is well written and intuitive. Further studies of generalisation in robot learning are overall highly relevant and this study contributes to better understanding. That being said, it seems to be very close to prior existing work Dittadi et al 2021 (citation in paper) and some insights are close to common knowledge.

The evaluation of generalisation also remains limited. After discussing the 7 different factors of variation in the domain, the full investigation purely focuses on color which limits the meaningfulness of the evaluation.

On the positive side, it’s great to see the focus on investigating proxy metrics which could be evaluated as a replacement to full reinforcement learning and this part of the paper is highly relevant.

It’s interesting that a sparsity inducing regularisation on the first policy layer has no effect on the usefulness of disentangled representations. Maybe it would be useful here to evaluate with sparsity regularisation across the whole policy.

Minor:
The initial section only cites papers starting 2019 regarding the problem of generalisation. This question has a much longer history in ML research which should be taken into account.
Table 1 on page 2 includes acronyms which are explained many pages later. Please make sure that an acronym is explained before using it on its own.
Figure 1 likely presents representative view points and not agent inputs. Additionally providing these would be more helpful in understanding the sim to real gap.
Why is there no OOD1 in real?
Please add an argument for the specific choice of the 2 types of VAEs. What is the benefit of adding Ada-GVAE in addition to beta-VAE?
Why not evaluate with beta=0 to include the regular autoencoder case?
The dimensionality of the latent space could have a big impact. I’d recommend at least a minor ablation with the best known parameters and sweeping over the size of the latent space.
Why are the correlations ‘milder’ on the more complex pushing task? It does not seem directly intuitive and it would be useful to add a hypothesis.
Some related work on representation learning for reinforcement learning is missing [1,2,3]

[1] Kulkarni, Tejas D. et al. “Unsupervised Learning of Object Keypoints for Perception and Control.” ArXiv abs/1906.11883 (2019): n. pag.
[2] Wulfmeier, Markus, et al. "Representation matters: Improving perception and exploration for robotics." 2021 IEEE International Conference on Robotics and Automation (ICRA). IEEE, 2021.
[3] Higgins, Irina et al. “Towards a Definition of Disentangled Representations.” ArXiv abs/1812.02230 (2018): n. pag.


**Summary Of The Paper:**

The paper discusses a study of different learned autoencoder based representations in the context of generalisation (in and out of distribution) in reinforcement learning. It focuses on two types of VAE models and a robotic block reaching/pushing setting. Comparisons are performed to separately test OOD data for the representation and learned policy as well as generalisation to a real world setting after training in simulation. The analysis shows correlations between various properties and metrics and final downstream RL performance



**Summary Of The Review:**

Overall a relevant investigation into the effect of representations on generalisation in RL. While some insights are trivial and the only investigated factor of variation for generalisation is color, there are aspects in the paper worth distributing and discussing (in particular the evaluation of proxy metrics for downstream performance).

---

> ### Author Response · Authors · 2021-11-15
> **Response to Reviewer AwPM. Part 1.**
>
>
> We thank the reviewer for their time and valuable feedback, and are encouraged by the generally positive review. We are glad to see that the reviewer agrees that “studies on generalisation in robot learning are highly relevant” and found our paper “well written and intuitive”. We are happy to see they considered our investigation on “proxy metrics which could be evaluated as a replacement to full reinforcement learning [...] highly relevant.” We address the reviewer’s comments in the following.
>
> &nbsp;
>
> **It seems to be very close to prior existing work (Dittadi et al. 2021).**
> The setup is by intention exactly the same as in Dittadi et al. but the focus and research question are entirely different: we tackle the generalization of downstream RL policies, whereas Dittadi et al. introduce the image dataset for representation learning and generalization, and conduct a preliminary study at the representation level only.
>
> **Some insights are close to common knowledge.**
> Although some of the results in this work might not be particularly surprising, we believe that even expected or intuitive results should be thoroughly validated in the scientific discourse.
>
> **The investigation purely focuses on color which limits the meaningfulness of the evaluation.**
> We want to clarify that we do not only focus on generalization w.r.t. object color: we also test on a novel shape and on the real robot, which are OOD w.r.t. the encoder. In addition, there are 6 factors (cube position, rotation, and joint angles) on which we cannot perform a similar investigation as with color, since we cannot realistically train RL agents while holding out some values of these factors.
>
> ***“Figure 1 likely presents representative view points and not agent inputs. Additionally providing these would be more helpful in understanding the sim to real gap.”***
> In panels 2 and 3 (a-c) we wanted to show the entire robotic setup for the sake of clarity, but we agree we should have highlighted that this is not the actual agent input. The agent input in simulation can be seen in panel 1, and the real world input is in panel 3d. Figure 1 in Dittadi et al. shows a direct sim-to-real comparison. We will add a similar comparison in the appendix for convenience, and would be happy to rework Figure 1 (e.g. by showing the agent input in all panels) for the camera-ready version.
>
> ***"Why is there no OOD1 in real?"***
> This is because the representation is purely trained in simulation. Thus, real-world observations are by definition OOD2. One can indeed imagine the following OOD1 scenario in the real-world: Train representations on real-world images under 8 different cube colors, then train a downstream predictor on four of these colours, and measure its performance on the other four colours. As Dittadi et al., we only consider training representations in simulation, because it is expensive to collect a large labeled dataset on the real robot.
>
> **Why did we choose AdaGVAE and beta-VAE, and what is the benefit of including Ada-GVAE?**
> We wanted to get a wide range of disentanglement across all studied representations. Beta-VAE is a relatively standard VAE approach, but cannot learn fully disentangled representations on this dataset. On the other hand, it leads to a diverse range of representations, also from a disentanglement perspective. Ada-GVAE is a weakly supervised counterpart of beta-VAE, and can yield almost perfect disentanglement scores in this scenario.
>
> **Why not evaluate with beta=0, i.e. the regular autoencoder case?**
> We wanted to match as closely as possible the common design choices in state-of-the-art methods. Beta=0 is typically not chosen in beta-VAEs since the goal is to encourage disentanglement by regularizing the KL from the prior to the approximate posterior even more strongly than in standard VAEs.

---

> > ### Author Response · Authors · 2021-11-15
> > **Response to Reviewer AwPM. Part 2.**
> >
> >
> > ***“The dimensionality of the latent space could have a big impact.”***
> > Although we initially considered this, we chose to limit this axis of our study because the latent space dimensionality affects the input size of the policy MLPs. If this were to be changed, it might be necessary to adjust the policy’s hyperparameters, thereby potentially introducing additional confounders in our study. On the other hand, we believe this not to be a particularly crucial parameter, provided it is not too small: We know exactly how many factors are needed to reconstruct the scene, and all representations are clearly expressive enough as they achieved excellent reconstruction. Adding more latents would therefore not necessarily help. We already observed “inactive latents” with the current latent size.
> >
> > ***“Why are the correlations ‘milder’ on the more complex pushing task?”***
> >  Our hypothesis is that this is due to the more complicated nature of this task. Solving pushing also requires learning the nonlinear dynamic rigid-body interactions between finger and cube. RL can strongly depend on the random seed and if a policy does not learn these mechanisms reliably it has trouble achieving high success scores, irrespective of the generalization capability of the representation, and has thus lower correlations. Note the higher variance of pushing policies in Figure 2.
> >
> > **Adding further related work.**
> > Thanks for the suggestion. These are indeed relevant and we added them to the current revision.

---

> > > ### Comment · Reviewer_AwPM · 2021-11-24
> > > **Feedback**
> > >
> > > Thank you very much for the additional feedback.
> > > The positions of the authors with respect to many questions have been clarified. Important ablations remain missing (latent dimensionality, no KL regularsation/regular AE) but the submission is relevant; bringing together some known perspectives and taking small steps towards extending our knowledge of generalisation properties of representation. Due to similarity to (Dittadi et al. 2021) and its incremental nature, my score will remain the same.

---

### Official Review · Reviewer_vE69 · 2021-11-02

**Correctness:** 4
**Technical Novelty And Significance:** 3
**Empirical Novelty And Significance:** 3
**Recommendation:** 8
**Confidence:** 3

**Main Review:**

Primary confusion/clarity concern: I had trouble understanding the rank-correlation results.  I think the clarity issues begin at the following quote in Section 2: “As factor prediction mode...as GS-OOD1, GS-OOD2-sim, and GS-OOD2-real accordingly.”  After reading these 4 sentences many times, I still think they are very difficult to understand (primarily on my first and second readthrough, but even now that I think I mostly understand the approach, I still think these sentences are confusing).  Similarly, sentences like “A simple supervised metric to evaluate a representation is how well a small downstream model can predict the ground-truth factors of variation” and “we use the MLP10000 and GBT10000 metrics (simply MLP and GBT in the following), where MLPs and GBTs are trained to predict the FoVs from 10,000 samples” and “rank correlations between representation metrics and training reward” need to be explained better.  (If space is a factor, then pointing to an appendix for more detailed/intuitive explanations is a good strategy.)

Minor problem: “Correlations with p<0.05 are color-coded.”  How are they color-coded?  This is probably obvious to a reader without any of the primary confusion above (I think it’s red, but even then, which shade of red?), but making it explicit would be better.

Minor grammar edit: “As factor prediction model, we will likewise...” missing article adjective?

Strengths:
- Outstanding limitations discussion; this frank discussion of limitations strengthens the paper.
- Thorough overall results with some fascinating insights.
- The sim-to-real experiments were very nice.

Questions:
- MLP stands for multi-layer perceptron, correct?  This should be defined.  GLB should be defined.
- What are the differences between MLP vs MLP(1-7)?
- Can you give any intuition about why MLP(1) might have a negative correlation in Figure 2?  Is this just noise?

**Update**:  After reading the authors' responses and other reviews, most of my concerns have been addressed (assuming the two changes I suggest are made).  I am updating my score to an 8; I feel that this is a strong paper. Regarding other reviewers' concerns:

I understand reviewer BEPU's concerns and agree with some of them; but do not think they are deal-breakers.  The paper has its limitations, but I think it is extremely strong despite these.  Robotics papers do tend to have more limitations than papers based exclusively on simulated/game environments, since real-world robotics experiments are much more time-consuming to perform (but they are also often much more convincing, since they are much closer to real-world problems of interests, and not games or simplistic simulations).

I respectfully disagree with many of Reviewer YvcV's concerns, many of which amount to "the authors should have done [some other nice thing]".  I think we can always ask authors to do more, and think the many of the reviewer's suggestions would be nice to do, but I think the contribution is sufficient in this case, and that rejecting because the authors did not do even more is not the correct response.  This paper's results are impressive and extensive despite the limitations that Reviewer YvcV points out.  I think part of the disconnect may be non-robotics RL researchers forgetting how time-consuming and worthwhile real-world robotics experiments are (compared to simulated or game experiments).

**Summary Of The Paper:**

The authors study the problem of generalizing out of distribution for RL.  They conduct an extensive empirical study, using simulation and real robotics, and observed properties that predict how well an agent will generalize.


**Summary Of The Review:**

This empirical paper has some clarity issues, but is exceptionally strong otherwise.

---

> ### Author Response · Authors · 2021-11-15
> **Response to Reviewer vE69**
>
> We thank the reviewer for their time and useful feedback. We are pleased to see the reviewer considers our work “exceptionally strong” and found the results “thorough with some fascinating insights” with “sim-to-real experiments [that] were very nice.” We are grateful for the constructive comments on clarity, which we address below. We would very much appreciate it if the reviewer would consider adjusting the score, given the otherwise very positive review.
>
> &nbsp;
>
> **Clarifications on evaluation metrics at the end of Section 2.**
>
> - *“As factor prediction mode...as GS-OOD1, GS-OOD2-sim, and GS-OOD2-real accordingly.”*: We kindly ask the reviewer to point us to the unclear aspects or ask specific questions regarding this quote. We would happily rephrase or add clarifying information accordingly.
>
> - *“A simple supervised metric to evaluate a representation is how well a small downstream model can predict the ground-truth factors of variation”*: Following previous work (Locatello 2019a&b, Dittadi et al 2021), we measure the usefulness of a representation in terms of how accurately a downstream model can predict the ground truth factors. If something is still unclear, we are happy to clarify and discuss further.
>
> - *“we use the MLP10000 and GBT10000 metrics (simply MLP and GBT in the following), where MLPs and GBTs are trained to predict the FoVs from 10,000 samples”*: If the reviewer finds it helpful, we would be happy to report the definitions of these metrics from Dittadi et al. 2021 in our background section as well.
>
> - *“rank correlations between representation metrics and training reward”*: This refers to correlations between each of the metrics discussed in our paper (unsupervised metrics, downstream factor prediction, generalization metrics, disentanglement metrics) and the average reward accumulated by an agent in one episode. Rank correlations measure monotonic relations between variables (as opposed to e.g. the Pearson correlation which measures linear relationships).
>
>
> **Color-coding of correlations.** Correlations with p-value > 0.05 are grey. All other correlations are red if positive and blue if negative, with intensity proportional to the correlation strength. We made this more explicit in the new revision.
>
> **Questions**:
>
> - **MLP stands for multi-layer perceptron, correct? This should be defined. GLB should be defined.** MLP indeed stands for Multi-Layer Perceptron. We never used the acronym GLB in our paper, but we guess the reviewer meant GBT, which stands for Gradient Boosted Trees. We defined these acronyms in the new revision.
>
> - **What are the differences between MLP vs MLP(1-7)?** MLP indicates the MLP metric from Dittadi et al. To evaluate this metric, we train 7 MLPs, one per FoV, to predict the ground-truth factors from the representation. The MLP metric is the average prediction error (on a held-out test set) of these 7 MLPs. MLP(_i_) denotes the prediction error on factor _i_ only. The meaning of the MLP(_i_) metrics and the numbering from 1 to 7 is explained in Figure 2 and in the caption of Figure 4.
>
> - **Can you give any intuition about why MLP(1) might have a negative correlation in Figure 2? Is this just noise?** We believe this to be noise, since it is a very mild correlation and the p-value is larger than 0.05 (it is grayed out).
>
> &nbsp;
>
> **References**
>
> Locatello et al. 2019a, Challenging Common Assumptions in the Unsupervised Learning of Disentangled Representations. ICML 2019.
>
> Locatello et al. 2019b, On the fairness of disentangled representations. NeurIPS 2019.
>
> Dittadi et al. 2021, On the transfer of disentangled representations in realistic settings. ICLR 2021.

---

> > ### Comment · Reviewer_vE69 · 2021-11-19
> > **Thank you**
> >
> > Thank you for the detailed reply; I understand much better now and will update my review accordingly.  I suggest two small changes:
> >
> > "the average reward accumulated by an agent in one episode" unless if I missed it, this is not stated clearly in the paper (I was thinking it might be something else, such as the return in the final few episodes of each run), so please consider adding this detail where appropriate (maybe the caption of Figure 2?).
> >
> > "To evaluate this metric, we train 7 MLPs, one per FoV, to predict the ground-truth factors from the representation. The MLP metric is the average prediction error (on a held-out test set) of these 7 MLPs. MLP(i) denotes the prediction error on factor i only. The meaning of the MLP(i) metrics and the numbering from 1 to 7 is explained in Figure 2 and in the caption of Figure 4."  This makes a lot of sense.  Unless if I missed it, I think a lot of this info is sort of implied but not stated this clearly; please try to state this info more explicitly (your explanation to me is perfect!).  Perhaps in the caption of Figure 2?

---

> > > ### Author Response · Authors · 2021-11-24
> > > **Thank you**
> > >
> > > We are very grateful to the reviewer for raising the score and for taking the time to read and comment on the other reviews.
> > >
> > > Although we cannot update the paper anymore, we will implement the 2 additional suggestions as follows:
> > > - We will include an explicit definition of how we evaluate the policies in the beginning of the experimental section.
> > > - We will more clearly define the MLP and MLP(i) metrics in Section 2, where the MLP metric is introduced.

---

### Comment · Area_Chair_Espb · 2021-11-28
**degree of OOD**

It would be great if the other reviewers could comment on reviewer vE69's updated review, in particular the point about whether there is a need to go farther out of distribution. I'd like to split into two questions:

* The OOD settings in the paper are clearly different from the training distribution, but just as clearly, one could design other settings with even higher distances from the training distribution. Given the difficulty of robotic experiments and sim2real transfer, is it enough of a contribution to consider the given degree of OOD-ness?
* Suppose that the degree of difference is sufficient. Within this framework, do the authors do a good job of justifying their claims?

It would be great if we could come to some kind of consensus about these questions &mdash; though of course it's better to continue to disagree than to force a false consensus. Thanks for the discussion!

---

### Decision · Program_Chairs · 2022-01-20

**Decision:**

Accept (Poster)

**Comment:**

It can be prohibitively expensive to train a reinforcement learner from scratch &mdash; particularly in cases where experience is expensive to obtain, such as with a physical robot. So, we might hope to speed up RL in a couple of ways: first, by pre-training a representation that makes subsequent RL need less data; and second by running our RL on a cheaper proxy environment such as a simulator. For pre-training, we hope to be able to take advantage of available pre-collected data, and we hope to be able to use supervised learning or reconstruction tasks since they can be cheaper than RL. For either pre-training or a proxy environment, we have to deal with distribution shifts: the properties of the environment may change between pre-training and RL, and between RL and testing the learned policy.

The paper presents an empirical study of how different pre-trained representations and different distribution shifts affect RL performance. It evaluates a number of representations trained by different VAEs (differing in aspects such as loss and hyperparameter settings) under various scenarios of distribution shift. It also asks whether we can predict the performance of the learned policies from properties of the representations, before going to the expense of training and evaluating our reinforcement learner.

The paper concludes that it is possible to significantly reduce RL data requirements using pre-trained representations, even in the presence of significant distribution shifts &mdash; including demonstrating zero-shot sim2real transfer. And, the paper concludes that inexpensive measurements of OOD performance on supervised tasks can at least partially predict success in generalization.

The reviewers praised the extensive experimental evaluation, including a large number of experiments on a physical robot, as well as the investigation of less-expensive ways to predict generalization.

Some reviewers were concerned that the choice of environments was limiting &mdash; e.g., that the distributional distance between in-distribution and out-of-distribution tests was limited, or that the results might not generalize to other related robotic environments. However, in the end there was support for the conclusion that the experiments cover a sufficiently general and interesting question.